# Direct synthesis of cyanate anion from dinitrogen catalysed by molybdenum complexes bearing pincer-type ligand

Takayuki Itabashi[1], Kazuya Arashiba[1], Akihito Egi[2], Hiromasa Tanaka[3], Keita Sugiyama[1], Shun Suginome [1], Shogo Kuriyama [1], Kazunari Yoshizawa [2] ✉ & Yoshiaki Nishibayashi [1] ✉

Dinitrogen is an abundant and promising material for valuable organonitrogen compounds containing carbon–nitrogen bonds. Direct synthetic methods for preparing organonitrogen compounds from dinitrogen as a starting reagent under mild reaction conditions give insight into the sustainable production of valuable organonitrogen compounds with reduced fossil fuel consumption. Here we report the catalytic reaction for the formation of cyanate anion ($NCO^-$) from dinitrogen under ambient reaction conditions. A molybdenum–carbamate complex bearing a pyridine-based 2,6-bis(di-*tert*-butylphosphinomethyl)pyridine (PNP)-pincer ligand is synthesized from the reaction of a molybdenum–nitride complex with phenyl chloroformate. The conversion between the molybdenum–carbamate complex and the molybdenum–nitride complex under ambient reaction conditions is achieved. The use of samarium diiodide ($SmI_2$) as a reductant promotes the formation of $NCO^-$ from the molybdenum–carbamate complex as a key step. As a result, we demonstrate a synthetic cycle for $NCO^-$ from dinitrogen mediated by the molybdenum–PNP complexes in two steps. Based on this synthetic cycle, we achieve the catalytic synthesis of $NCO^-$ from dinitrogen under ambient reaction conditions.

Dinitrogen is an abundant and promising material for valuable organonitrogen compounds containing carbon–nitrogen bonds. In general, it is considerably difficult to convert dinitrogen into valuable organonitrogen compounds because of the thermodynamic and kinetic stability of dinitrogen[1]. To use dinitrogen as a nitrogen source, dinitrogen is converted into ammonia by the Haber–Bosch process, which requires harsh reaction conditions and consumes over 1% of world fossil fuel[2]. In addition, some steps are necessary to produce valuable organonitrogen compounds from ammonia. Therefore, direct synthetic methods for preparing organonitrogen compounds from dinitrogen as a starting reagent under mild reaction conditions

give insight into the sustainable production of valuable organonitrogen compounds with reduced fossil fuel consumption.

To overcome the inertness of dinitrogen, the direct conversion of dinitrogen into organonitrogen compounds mediated by transition metal complexes has been conducted for the last half-century[3–6]. In particular, organonitrogen compounds can be prepared from the reactions of transition metal–dinitrogen complexes with carbon-centred electrophiles to functionalise the coordinated dinitrogen on the metal atoms under mild reaction conditions[5–11]. Since the milestone work for the cleavage of dinitrogen to synthesise the corresponding nitride complex reported by the Cummins' group[12,13], organonitrogen

[1]Department of Applied Chemistry, School of Engineering, The University of Tokyo, Bunkyo-ku, Tokyo 113-8656, Japan. [2]Institute for Materials Chemistry and Engineering, Kyushu University, Nishi-ku, Fukuoka 819-0395, Japan. [3]School of Liberal Arts and Sciences, Daido University, Minami-ku, Nagoya 457-8530, Japan. ✉e-mail: kazunari@ms.ifoc.kyushu-u.ac.jp; ynishiba@g.ecc.u-tokyo.ac.jp

compounds can be synthesised from the reactions of transition metal–nitride complexes. This is an alternative method to prepare organonitrogen compounds from dinitrogen. The transition metal–nitride complexes, which are generated via the direct cleavage of a nitrogen–nitrogen triple bond, reacted with carbon-centred electrophiles to functionalise the nitride ligand under mild reaction conditions[14–26]. Based on the experimental results of stoichiometric reactions with transition metal complexes, several research groups have constructed synthetic cycles to prepare organonitrogen compounds from dinitrogen by step-by-step reactions requiring a stoichiometric amount of transition metal complexes[9–11,14–19,22,23,25–27]. However, the catalytic and direct transformations of dinitrogen into valuable organonitrogen compounds with transition metal complexes under mild reaction conditions have not been reported until now.

As the first investigation for the catalytic and direct preparation of organonitrogen compounds from dinitrogen, we have focused on isocyanates (R–N=C=O) and their derivatives using CO and its derivatives because isocyanate derivatives are attractive synthetic compounds from the viewpoint of thermodynamics and usefulness in organic synthesis[28]. In particular, sodium and potassium salts of isocyanate are commercially important with the world demand of 8000–10000 tonnes per year. The isocyanate salts are applied in steel hardening and fine chemical and pharmaceutical synthesis as well as in the agrochemical industry. Currently, the isocyanate salts are produced from the reaction of the corresponding carbonate salts with urea at over 400 °C, which overall requires two moles of $NH_3$ for each mole of cyanate[29]. Recently, some research groups have achieved synthetic cycles to prepare isocyanate derivatives from dinitrogen via stepwise procedures[22,23,25,26]. Kawaguchi's group and Schneider's group have reported synthetic cycles for isocyanate derivatives in five and seven steps, respectively (Fig. 1a, b)[22,25]. Both synthetic cycles comprise the formation of the corresponding isocyanate complexes from the reactions of the nitride complexes, which are formed via the direct cleavage of a nitrogen–nitrogen triple bond of dinitrogen, with carbon monoxide (CO) as key reaction steps. As a related work, Sita's group has reported a synthetic cycle for trimethylsilylisocyanate in five steps, where the reaction of a molybdenum–silylimide complex, which is formed via the direct cleavage of a nitrogen–nitrogen triple bond of dinitrogen and silylation, with $CO_2$ is a key reaction step (Fig. 1c)[23]. In these cases, it is considerably difficult to expand the synthetic cycles into the corresponding catalytic cycles. This is because of the multiple (five and seven) steps under different reaction conditions and the use of CO and $CO_2$ to inhibit the generation of the metal–nitride complexes via the cleavage of dinitrogen. Quite recently, Hou's group has reported a synthetic cycle for trimethylsilylisocyanate in four steps, where the reaction of titanium–dinitrogen complex with $CO_2$ is a key step (Fig. 1d)[26]. However, for the same reason, it is also difficult to achieve the corresponding catalytic reaction.

Recently, we have found effective catalytic ammonia formation from dinitrogen under ambient reaction conditions. A molybdenum–nitride complex bearing a pyridine-based PNP-type pincer ligand [Mo(N)I(PNP)] (1) (PNP = 2,6-bis(di-*tert*-butyl-phosphinomethyl)pyridine), which is formed via the direct cleavage of a nitrogen–nitrogen triple bond of dinitrogen, works as a key reactive intermediate (Fig. 2a)[30–32]. In addition, we have investigated the reactions of 1 with various carbon-centred electrophiles towards preparing organonitrogen compounds[33]. The stoichiometric amount of organic amide is successfully synthesised from the reaction of 1 with acyl chloride and water.

In this work, we report the synthesis of cyanate anion (NCO−) from dinitrogen using chloroformate ester mediated by molybdenum complexes under ambient reaction conditions (Fig. 2b). Some research groups previously used chloroformate esters as a carbon monoxide source to obtain the corresponding ketones from transition metal-catalysed reactions[34,35]. The synthetic cycle reported herein comprises

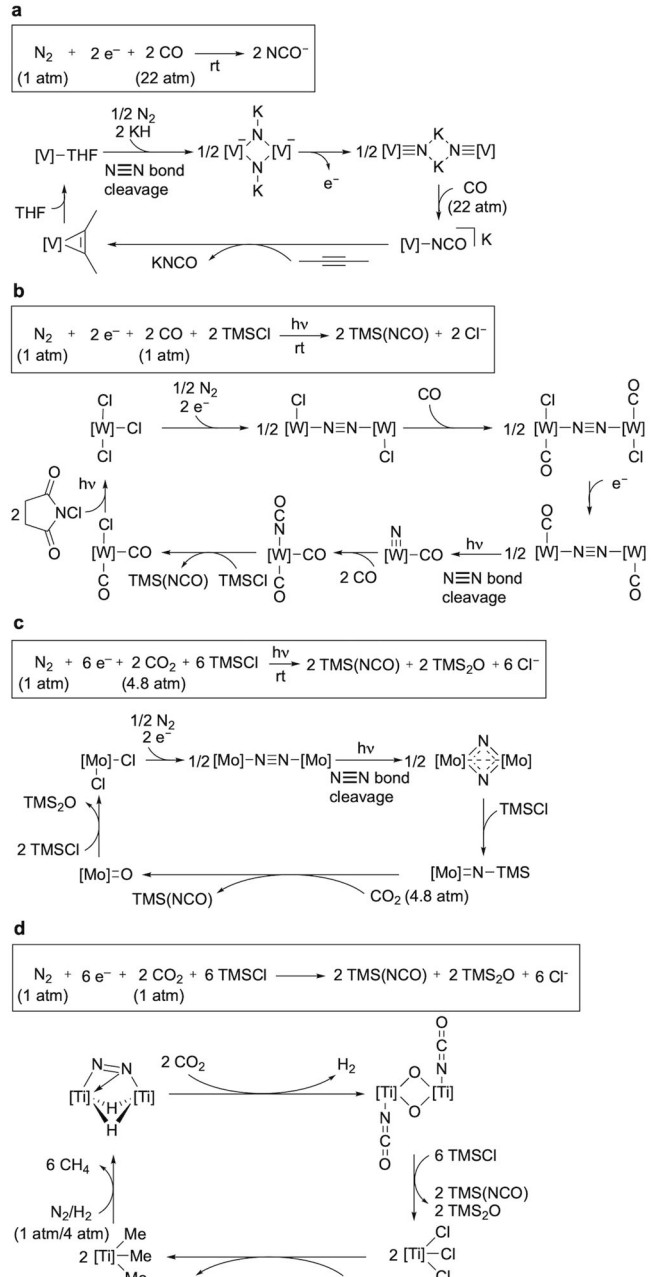

**Fig. 1 | Synthetic cycles for isocyanate derivatives from dinitrogen under mild reaction conditions. a** Previous work 1: Kawaguchi's work. **b** Previous work 2: Schneider's work. **c** Previous work 3: Sita's work. **d** Previous work 4: Hou's work.

the carbamylation of the molybdenum–nitride complex and the reduction of the molybdenum–carbamate complex followed by the direct cleavage of a nitrogen–nitrogen triple bond of dinitrogen in two steps. This methodology enables the catalytic production of NCO− from dinitrogen under ambient reaction conditions.

## Results
### Synthetic cycle for NCO− from dinitrogen
The reaction of 1 with 1.1 equiv of phenyl chloroformate in tetrahydrofuran (THF) at room temperature for 2 h afforded the corresponding carbamate complex [Mo(NCO₂Ph)ICl(PNP)] (2) in 83% yield (Fig. 3a). This carbamate complex 2 was characterised by ¹H and ³¹P{¹H} nuclear magnetic resonance (NMR). The presence of a carbamate ligand was supported by infrared (IR) measurement (1690 cm⁻¹ ($v_{C=O}$)

**a**

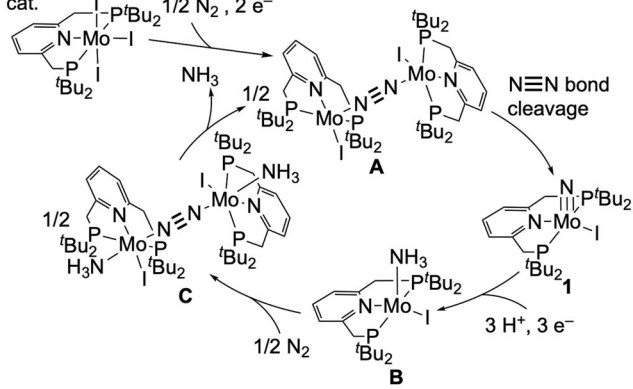

**proposed catalytic cycle**

**b**

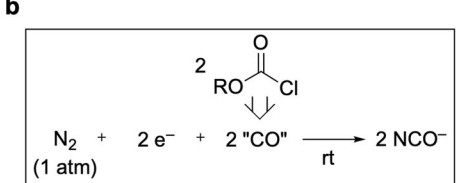

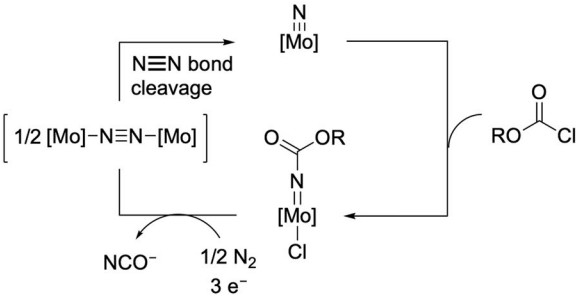

**Fig. 2 | Catalytic reactions from dinitrogen under ambient reaction conditions.**
**a** Previous work: catalytic ammonia formation via the direct cleavage of dinitrogen;
**b** This work: catalytic cyanate anion formation via the direct cleavage of dinitrogen.

and 642 cm$^{-1}$ ($\nu_{Mo=N}$)). According to the same method, the corresponding $^{15}$N-labelled carbamate complex [Mo($^{15}$NCO$_2$Ph)ICl(PNP)] (**2-$^{15}$N**) was obtained from the reaction of the $^{15}$N-labelled nitride complex [Mo($^{15}$N)I(PNP)] (**1-$^{15}$N**), which was prepared from the reaction of [MoI$_3$(PNP)] with 2.2 equiv of KC$_8$ under 1 atm of $^{15}$N$_2$, with phenyl chloroformate. This complex **2-$^{15}$N** was characterised by $^1$H, $^{31}$P{$^1$H}, and $^{15}$N{$^1$H} NMR. The $^{15}$N{$^1$H} NMR of **2-$^{15}$N** showed a resonance at −51.6 ppm (t, $J_{N-P}$ = 4.9 Hz). Interestingly, the reaction of **1** with 1.1 equiv of phenyl iodoformate under similar reaction conditions did not afford the corresponding carbamate complex but the corresponding cationic nitride complex [Mo(N)I(PNP)][I] (**[1][I]**) quantitatively (see Supplementary Information). This result indicates that the use of chloroformate as a carbamylation reagent is necessary to obtain the corresponding carbamate complex.

The detailed molecular structure of carbamate complex **2** was confirmed by X-ray crystallography. An ORTEP drawing of **2** is shown in Fig. 3b. Complex **2** has a distorted octahedral geometry around the molybdenum atom, where a carbamate ligand is coordinated to the molybdenum centre with the Mo1-N2 bond distance of 1.766(3) Å and the Mo1-N2-C24 angle of 173.4 (2)° in the apical position to the nitrogen atom of the pyridine ring. An iodo ligand and a chloro ligand are

presented in *cis* and *trans* positions to the carbamate ligand, respectively. The bond distances and angles of **2** are similar to that of an acylimide complex [Mo(NCOCH$_2$Ph)ICl(PNP)][33].

The reduction of carbamate complex **2** with 5 equiv of SmI$_2$ in THF at room temperature for 2 h under an atmospheric pressure of dinitrogen afforded nitride complex **1** and samarium phenoxy complex [SmI$_2$(OPh)] (**3a**) in 74 and 93% NMR yields, respectively. Further, samarium isocyanate species, such as [SmI$_2$(NCO)], were formed as a formal composition formula (Fig. 3d). Samarium complex **3a** was separately prepared from the reaction of SmI$_3$ by ligand exchange with 1 equiv of KOPh, and the molecular structure of **3a** was confirmed by preliminary X-ray crystallography (see Supplementary Information). Unfortunately, the molecular structure of the samarium isocyanate species has not been characterised until now. The amount of the formed NCO$^-$ after the hydrolysis of the reaction mixture was determined by ion chromatography (IC), where 0.75 equiv of NCO$^-$ were obtained based on **2** (75% IC yield). The formation of **1** and NCO$^-$ was characterised by electrospray ionisation–mass spectroscopy (ESI–MS) ($m/z$ = 634.18) and IR ($\nu_{14NCO}$ = 2170 cm$^{-1}$), respectively.

Instead of the characterisation of the samarium isocyanate species, such as [SmI$_2$(NCO)], a samarium isocyanate complex {[SmI(NCO)(18-crown-6)][I]}$_n$ (**3b**) (18-crown-6 = 1,4,7,10,13,16-hexaoxacyclooctadecane) was separately prepared in 60% yield from the reaction of SmI$_2$ with 1 equiv of AgNCO as an oxidant with 1 equiv of 18-crown-6. The molecular structure of **3b** was confirmed by preliminary X-ray crystallography (see Supplementary Information). The hydrolysis of **3b** quantitatively afforded NCO$^-$. Therefore, we suppose that the samarium isocyanate species [SmI$_2$(NCO)] also quantitatively released NCO$^-$ after hydrolysis.

The transformation of carbamate complex **2** into nitride complex **1** and NCO$^-$, as shown in Fig. 3d, is an unexpected and interesting result. To trace the origin of the N-atom in **1** and NCO$^-$ in this reaction, an isotope labelling experiment was conducted (Fig. 3e). The reduction of $^{15}$N-labelled carbamate complex [Mo($^{15}$NCO$_2$Ph)ICl(PNP)] **2-$^{15}$N** with 5 equiv of SmI$_2$ in THF at room temperature for 2 h under an atmospheric pressure of $^{14}$N$_2$ gas afforded nitride complex [Mo($^{14}$N)I(PNP)] **1** and [SmI$_2$(OPh)] **3a** in 41 and 97% NMR yields, respectively, with $^{15}$N-labelled cyanate anion ($^{15}$NCO$^-$) in 62% IC yield. The formation of [Mo($^{14}$N)I(PNP)] **1** and $^{15}$NCO$^-$ was confirmed by ESI–MS ($m/z$ = 634.18) and IR ($\nu_{15NCO}$ = 2153 cm$^{-1}$), respectively. This isotope labelling experiment suggests that, in this reaction system, [{MoI(PNP)}$_2$(μ-N$_2$)] (**A**, shown in Fig. 2a) was formed after the dissociation of $^{15}$N-labelled cyanate anion $^{15}$NCO$^-$ from the corresponding molybdenum complex, followed by the reaction with nitrogen gas ($^{14}$N$_2$) to form the nitride complex via the direct cleavage of the bridging dinitrogen on the dinitrogen-bridged dimolybdenum complex.

The experimental results shown in Fig. 3a, d, e suggest that the conversion of dinitrogen into cyanate anion under ambient reaction conditions is possible via the interconversion between the nitride and carbamate complexes (Fig. 3f).

To get more information on the reaction pathway, the one-electron reduction of carbamate complex **2** was conducted. The reduction of **2** with only 1.0 equiv of SmI$_2$ in THF at room temperature for 10 min under an atmospheric pressure of Ar afforded the corresponding isocyanate complex [Mo(NCO)ICl(PNP)] (**4a**) in 68% yield (Fig. 3g). In the IR spectrum, an isocyanate (NCO) stretching vibration was observed at 2209 cm$^{-1}$. A redshift ($\Delta\nu_{15N}$ = 14 cm$^{-1}$) was observed in the $^{15}$N-labelled isocyanate complex [Mo($^{15}$NCO)ICl(PNP)](**4a-$^{15}$N**). This result indicates that the isocyanate complex was formed from the carbamate complex via carbon–oxygen bond cleavage in the carbamate ligand during the reductive conversion. In contrast, the reduction of **2** with 1.1 equiv of CoCp*$_2$ (Cp* = η$^5$-C$_5$Me$_5$) in THF at room temperature for 16 h under an atmospheric pressure of Ar afforded another isocyanate complex [Mo(NCO)(OPh)Cl(PNP)] (**4b**) in 49% yield (see Supplementary Information). These results indicate that the high

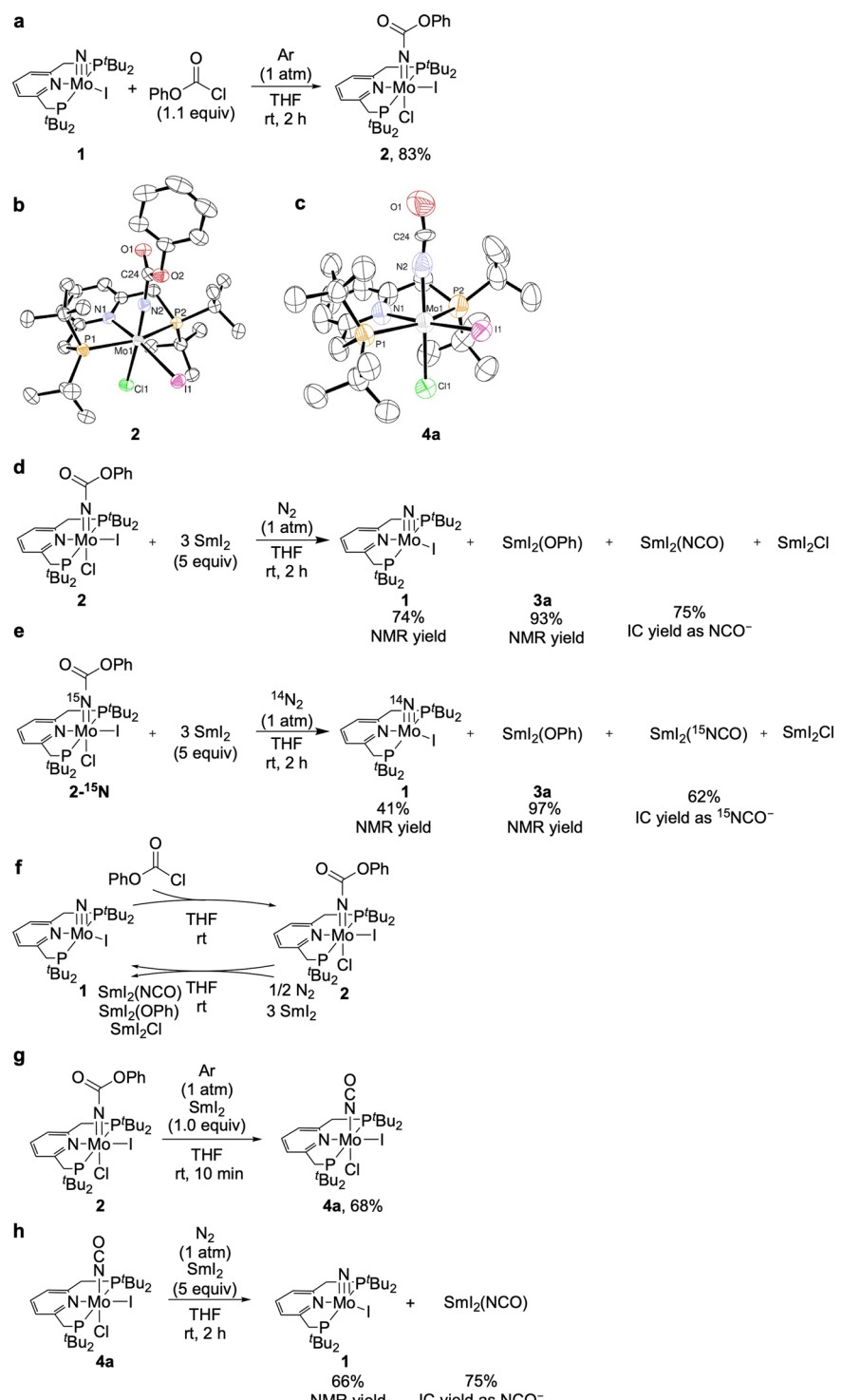

**Fig. 3 | Stoichiometric reactions for isocyanate derivatives from dinitrogen.**
**a** Synthesis of carbamate complex **2**. **b**, **c** ORTEP drawings of **2** (**b**) and **4a** (**c**). Thermal ellipsoids are shown at the 50% probability level. All hydrogen atoms and solvent molecules are omitted for clarity. **d** Reduction of **2** with an excess amount of SmI$_2$ under 1 atm of N$_2$. **e** Reduction of **2-$^{15}$N** with SmI$_2$ under 1 atm of $^{14}$N$_2$. **f** Two-step synthetic cycle for cyanate anion. **g** Synthesis of isocyanate complex **4a**. **h** Reduction of **4a** using an excess amount of SmI$_2$ under 1 atm of N$_2$.

oxophilicity of samarium species plays an essential role to remove the phenoxy ligand from the molybdenum complex.

An ORTEP drawing of **4a** is shown in Fig. 3c. Complex **4a** exhibited a distorted octahedral geometry around the molybdenum atom, where an isocyanate ligand was coordinated to the molybdenum centre in the apical position to the nitrogen atom of the pyridine ring. An iodo ligand and a chloro ligand are presented in *cis* and *trans*

positions to the isocyanate ligand, respectively. A detailed discussion of bond lengths and bond angles around the isocyanate ligand is difficult because of a slight disorder between the isocyanate and the chloro ligands.

The reduction of **4a** with 5 equiv of SmI$_2$ in THF at room temperature for 2 h under an atmospheric pressure of dinitrogen afforded the nitride complex [Mo(N)I(PNP)] **1** in 66% NMR yield, with cyanate

**Fig. 4 | A proposed reaction pathway for producing isocyanate derivatives from dinitrogen.** (i) The carbamylation of nitride complex **1** with phenyl chloroformate affords carbamate complex **2**. (ii) Reductive carbon−oxygen bond cleavage in the carbamate ligand of **2** produces isocyanate complex **4a**. (iii) After one-electron reduction, the dissociation of the chloride ligand of **4a** affords a five-coordinate isocyanate complex. A pair of the isocyanate complex sandwiches a dinitrogen molecule to form a dinitrogen-bridged dimolybdenum complex with a Mo−N≡N−Mo core, which is a precursor of [{MoI(PNP)}$_2$(μ-N$_2$)] **A**. (iv) The bridging N$_2$ ligand of **A** undergoes a direct N≡N bond cleavage to regenerate **1**.

anion NCO$^-$ in 75% IC yield (Fig. 3h). This result indicates that isocyanate complex **4a** is an intermediate in the transformation of carbamate complex **2** into nitride complex **1** and cyanate anion, as shown in Fig. 3d. Contrarily, the reduction of **4b** using 3 equiv of CoCp*$_2$ with 3 equiv of NaI in THF at room temperature under an atmospheric pressure of dinitrogen did not afford nitride complex **1** but some paramagnetic species (see Supplementary Information). These experimental results indicate that the use of SmI$_2$ as a reductant plays an essential role to transfer the isocyanate and chloro ligands from the molybdenum atom into the samarium atom. The removal of the isocyanate, phenoxy, and chloro moieties from **2** was necessary to regenerate nitride complex **1** via the direct cleavage of the bridging dinitrogen ligand on the dinitrogen-bridged dimolybdenum complex.

Based on these stoichiometric reactions shown in Fig. 3a, d–h, a detailed reaction pathway of the synthetic cycle for NCO$^-$ from dinitrogen mediated by molybdenum complexes can be proposed (Fig. 4). First, the carbamylation of nitride complex **1** with phenyl chloroformate affords the corresponding carbamate complex **2**. Next, the reduction of **2** with an excess amount of SmI$_2$ produces the dinitrogen-bridged dimolybdenum complex **A** and samarium complex, such as [SmI$_2$(OPh)] **3a**, [SmI$_2$(NCO)], and [SmI$_2$Cl], via the formation of the corresponding isocyanate complex **4a**. Finally, the dinitrogen-bridged dimolybdenum complex **A** is converted into the starting nitride complex **1** via the direct cleavage of the bridging dinitrogen ligand of **A**.

Thus, this synthetic cycle for the formation of NCO$^-$ from dinitrogen mediated by the molybdenum−PNP complexes was closed in two steps, carbamylation and reduction (Fig. 3f). The overall yield of NCO$^-$ in the reaction of dinitrogen with phenyl chloroformate and SmI$_2$ was 62% yield based on the molybdenum complex. This synthetic cycle provides a pathway for producing organonitrogen compounds from dinitrogen under mild reaction conditions with only two steps. The two steps are the fewest number of steps in the so-far reported synthetic cycles that consist of the combination of various stoichiometric reactions[22,23,25,26]. In addition, the formation of the isocyanate complexes from the reaction of the corresponding nitride complexes with CO was a key step for previously reported isocyanate derivatives synthesis from dinitrogen[22,25]. In contrast to these reports, the reaction

of nitride complex **1** with CO did not afford an isocyanate complex but a cationic nitride carbonyl complex[33]. Therefore, this stoichiometric reaction is noteworthy as an alternative synthetic method for isocyanate derivatives from dinitrogen.

## DFT calculations to support the proposed pathway

Here, we have performed density-functional theory calculations at the B3LYP-D3 level to theoretically examine the proposed synthetic cycle shown in Fig. 4. This synthetic cycle can be divided into four reaction steps (i)–(iv): (i) The carbamylation of nitride complex **1** with phenyl chloroformate affords carbamate complex **2**. (ii) Reductive carbon−oxygen bond cleavage in the carbamate ligand of **2** produces isocyanate complex **4a**. (iii) After one-electron reduction, the dissociation of the chloride ligand of **4a** affords a five-coordinate isocyanate complex. A pair of the isocyanate complex sandwiches a dinitrogen molecule to form a dinitrogen-bridged dimolybdenum complex with a Mo−N≡N−Mo core, which is a precursor of [{MoI(PNP)}$_2$(μ-N$_2$)] **A**. (iv) The bridging N$_2$ ligand of **A** undergoes a direct N≡N bond cleavage to regenerate **1**. We experimentally and computationally demonstrated that the direct N≡N bond cleavage of **A** is a key reaction step in the catalytic transformation of dinitrogen into ammonia using Mo−PNP complexes[30,32]. Therefore, we would like to propose that steps (iii) and (iv) should be involved in the stoichiometric transformation of dinitrogen into NCO$^-$. In this paper, we will focus on steps (i)–(iii), since step (iv) has been thoroughly investigated[30].

Figure 5a presents a free energy profile at 298 K ($\Delta G_{298}$) in THF calculated for step (i). The carbonyl carbon atom of phenyl chloroformate attacked the nitride ligand of **1**. The formation of the C−N bond was linked to the cleavage of the C−Cl bond (**RC** → **TS** → **PC**), where **RC**, **TS** and **PC** represent reactant complex, transition state and product complex, respectively. The released Cl$^-$ anion occupied the vacant coordination site of the molybdenum centre to afford **2**. The formation of **2** was highly exergonic by 36.6 kcal/mol, with a very low activation free energy of $\Delta G_{298}^{\ddagger} = 5.3$ kcal/mol.

Next, we move to step (ii) in Fig. 4. At this step, the reduction of **2** was followed by the C−O bond cleavage to yield isocyanate complex **4a** and a phenolate (PhO$^-$). Cyclic voltammetry of **2** showed an irreversible reduction wave at $E_{pc} = -1.76$ V vs. FeCp$_2^{0/+}$ (Cp = η$^5$-C$_5$H$_5$), which was assignable to the one-electron reduction of **2** (Supplementary Fig. 6). Therefore, our theoretical consideration started with the reduced species of **2** (complex **I** in Fig. 5b). The C−O bond cleavage of **I** smoothly occurred with a spin inversion between the doublet and quartet states. The formation of **4a** (quartet) from **I** (doublet) went through **TS$_{I/4a}$** (quartet), with a considerably low activation free energy of 1.6 kcal/mol, and proceeded in a highly exergonic way by 30.4 kcal/mol. During this reaction step, the Mulliken spin density was localised at the Mo centre, and no spin density was assigned to the leaving phenolate group (Supplementary Fig. 27). These results suggest that step (ii) can be regarded as the heterolysis of the C−OPh bond.

In step (iii), dimolybdenum complex **A** with a Mo$^I$−N≡N−Mo$^I$ core was formed from Mo$^{III}$ complex **4a** via two-electron reduction, the dissociation of both NCO and Cl ligands and the coordination of dinitrogen. First, the NCO or Cl ligand was dissociated from the six-coordinate Mo centre of **4a** because a vacant coordination site must be prepared for incoming dinitrogen. As presented in Fig. 6a, a Mo$^{II}$ complex **II**, the reduced species of **4a**, undergoes a facile dissociation of Cl$^-$ (**II** → **III** + Cl$^-$; $\Delta G_{298} = 1.1$ kcal/mol). Cyclic voltammetry of **4a** showed a quasi-reversible reduction wave at $E_{pc} = -1.86$ V vs. FeCp$_2^{0/+}$, assignable to the one-electron reduction of **4a** (Supplementary Fig. 7). The NCO ligand should remain coordinated because the $\Delta G_{298}$ value for the dissociation of NCO$^-$ from **II** (**II** → MoICl(PNP) + NCO$^-$) was calculated to be 11.8 kcal/mol. As described later, the release of the NCO group from the Mo centre will be accomplished by forming a dimolybdenum Mo−N≡N−Mo structure. Afterward, an incoming dinitrogen molecule is coordinated to a five-coordinate Mo−NCO complex

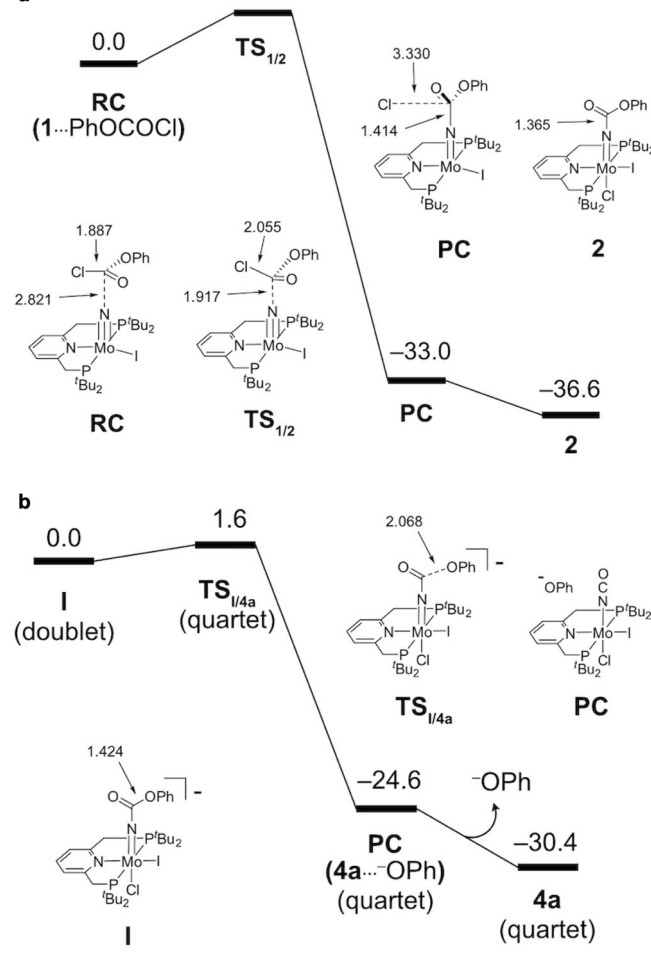

**Fig. 5 | A possible reaction pathway for the conversion of nitride complex 1 to isocyanate complex 4a. a** Free energy profile at 298 K ($\Delta G_{298}$) calculated for the reaction of nitride complex **1** with phenyl chloroformate yielding carbamate complex **2** in the closed-shell singlet state. Selected interatomic distances are presented in Å. **b** Free energy profile at 298 K ($\Delta G_{298}$) calculated for the formation of **4a** from **I** in the ground spin state (see more details in Supplementary Information). Selected interatomic distances are presented in Å.

**III**. The coordination of $N_2$ smoothly proceeded in an exergonic way by 9.4 kcal/mol (**III** + $N_2$ → **IV**). The generated complex **IV** reacted with another molecule of **III** to afford a six-coordinate dinitrogen-bridged dimolybdenum complex **V**, [{Mo$^{II}$(NCO)I(PNP)}$_2$(μ-N$_2$)]. The bridging dinitrogen strongly united two of the Mo$^{II}$–NCO units (**III** + **IV** → **V**; $\Delta G_{298}$ = −19.8 kcal/mol). We here assume a stepwise reduction of **V** towards the generation of the five-coordinate dimolybdenum complex **A**, as presented in Fig. 6b. The one-electron reduction of **V** caused a facile dissociation of one of the NCO ligands occupying the *trans* positions of the bridging dinitrogen (**VI** → **VII** + NCO$^-$; $\Delta G_{298}$ = −0.8 kcal/mol). The generated Mo$^I$–Mo$^{II}$ complex **VII** readily liberated the remaining NCO ligand after the one-electron reduction (**VIII** → **A** + NCO$^-$; $\Delta G_{298}$ = + 0.3 kcal/mol). The calculated results indicate that the formation of the Mo–N≡N–Mo core efficiently weakened the Mo$^I$–NCO bonds. We previously reported a similar situation in the reaction pathway for the formation of **A** from a dinitrogen-bridged Mo$^I$–Mo$^I$ complex **C**, [{Mo$^I$(NH$_3$)I(PNP)}$_2$(μ-N$_2$)], as shown in Fig. 2a[32]. In conclusion, from a theoretical point of view, the proposed synthetic cycle shown in Fig. 4 is reasonable if **4a** can undergo two-electron reduction.

Here we would like to discuss the role of samarium complexes in the proposed synthetic cycle. As described in the experimental

section, SmI$_2$ may play a role to transfer the phenoxy and/or the isocyanate groups from the molybdenum atom into the samarium atom. We computationally examined the possibility of two transfer reactions from a thermodynamic viewpoint; (a) the transfer of PhO$^-$ of [**4a**···$^-$OPh] to a cationic Sm(III) species [Sm$^{III}$I$_2$(THF)$_5$]$^+$, and (b) the transfer of NCO$^-$ of **VI** to [Sm$^{III}$I$_2$(THF)$_5$]$^+$. In the present calculations, we chose [Sm$^{III}$I$_2$(THF)$_5$]$^+$ as an acceptor of the leaving groups because the release of both PhO$^-$ and NCO$^-$ groups is preceded by the reduction of Mo complexes by SmI$_2$. The structure of the Sm(III) species in THF was taken from the crystal structure of SmI$_2$, [Sm$^{II}$I$_2$(THF)$_5$][36]. The $\Delta G_{298}$ values for the transfer reactions were calculated to be −26.2 kcal/mol for reaction [**4a**···$^-$OPh] + [SmI$_2$(THF)$_5$]$^+$ → **4a** + SmI$_2$(OPh)(THF)$_4$ + THF, and −10.5 kcal/mol for reaction **VI** + [SmI$_2$(THF)$_5$]$^+$ → **VII** + SmI$_2$(NCO)(THF)$_4$ + THF, both of which are highly exergonic. These results could support that the presence of the Sm(III) species inhibits undesired side reactions such as formation of Mo-OPh intermediates and promotes subsequent reactions.

## Catalytic formation of NCO$^-$ from dinitrogen

The two stoichiometric reactions in the synthetic cycle achieved above, carbamylation and reduction, can be carried out under almost the same reaction conditions. This result prompts us to investigate the catalytic reaction based on the stoichiometric reaction. Typical results are shown in Table 1. The reaction of the atmospheric pressure of dinitrogen with 36 equiv of SmI$_2$ as a reductant and 12 equiv of phenyl chloroformate as a carbon-centred electrophile with a catalytic amount of **1** in THF at room temperature for 16 h afforded 0.45 equiv of NCO$^-$ based on the molybdenum atom of the catalyst (Table 1, run 1). Next, we examined other molybdenum–nitride complexes bearing different pincer ligands. The use of a molybdenum–nitride complex bearing a PPP-type pincer ligand [Mo(N)Cl(PPP)] (**5**)[37] (PPP = bis(di-*tert*-butylphosphinoethyl)phenylphosphine) afforded a relatively low amount of NCO$^-$ (0.24 equiv based on the molybdenum atom) than that of **1**. However, the use of a molybdenum–nitride complex bearing a carbene-based PCP-type pincer ligand [Mo(N)I(PCP)] (**6**)[38] (PCP = 1,3-bis(di-*tert*-butylphosphinomethyl)benzimidazol-2-ylidene) as a catalyst slightly increased the amount of NCO$^-$ formed (0.74 equiv based on molybdenum atom of the catalyst; Table 1, runs 2 and 3). These results indicate that only a stoichiometric amount of NCO$^-$ was formed under these reaction conditions.

We envisaged that a direct reaction between SmI$_2$ and phenyl chloroformate may suppress the catalytic formation of NCO$^-$. Therefore, the slow addition of a THF solution containing phenyl chloroformate using a syringe pump to the mixture of **6** and SmI$_2$ in a THF solution was investigated as a next step. This modification of the procedure substantially increased the amount of NCO$^-$ to 1.43 equiv based on the molybdenum atom of the catalyst (Table 1, run 4). Furthermore, when methyl chloroformate was used instead of phenyl chloroformate to reduce the reactivity of chloroformate ester towards SmI$_2$, 2.51 equiv of NCO$^-$ were obtained based on the molybdenum atom of the catalyst (Table 1, run 5). This result indicates that a superstoichiometric amount of NCO$^-$ was formed from dinitrogen under ambient reaction conditions.

Contrarily, the use of ethyl chloroformate and isopropyl chloroformate was less effective than that of methyl chloroformate in this system (Table 1, runs 6 and 7). These results indicate that the bulkiness of the substituent on the chloroformate ester was critical for the present reaction system. We confirmed that catalyst, reductant, chloroformate ester and dinitrogen were essential factors to form a superstoichiometric amount of NCO$^-$ based on the molybdenum atom of the catalyst (Table 1, runs 8–11). Finally, the slow addition of 24 equiv of methyl chloroformate for an increased period, such as 12 h, yielded 8.99 equiv of NCO$^-$ based on the molybdenum atom of the catalyst (75% yield based on SmI$_2$; Table 1, run 12). This experimental result indicates that the catalytic formation of NCO$^-$ from dinitrogen

**Fig. 6 | A possible reaction pathway for the conversion of isocyanate complex 4a to dimolybdenum complex A. a** Free energy profiles at 298 K ($\Delta G_{298}$) calculated for the formation of **V** from **4a**. **b** Free energy profiles at 298 K ($\Delta G_{298}$) calculated for the dissociation of two equivalent of NCO⁻ from **V** to yield **A**.

was achieved under ambient reaction conditions. Thus, the amount of NCO⁻ from the catalytic reaction is 9 times greater than the theoretical amount of NCO⁻ produced from the stoichiometric reaction. As a result, we believe that this is the first successful example of the direct conversion of dinitrogen into organonitrogen compounds, such as NCO⁻, with transition metal complexes as catalysts under ambient reaction conditions.

## Discussion

We demonstrated a two-step synthetic cycle of cyanate anion NCO⁻ from dinitrogen with molybdenum complexes bearing a pyridine-based PNP-type pincer ligand under ambient reaction conditions. The reductive transformation of molybdenum–carbamate complex into molybdenum–isocyanate complex is a key step to promote the synthetic cycle. The overall yield of the formation of cyanate anion from the reaction of dinitrogen with SmI₂ as a reductant and phenyl chloroformate as a carbon-centred electrophile was 62% based on the molybdenum atom of the complex. This synthetic cycle is the fewest number of steps in the so-far reported stoichiometric reactions. Based on this synthetic cycle, we achieved the catalytic reaction for the

formation of NCO⁻ from dinitrogen under ambient reaction conditions. We believe that the results described in this manuscript provide valuable information to achieve the catalytic and direct transformations of dinitrogen into more valuable organonitrogen compounds under ambient reaction conditions.

## Methods
### General information

Detailed experimental procedures, characterization of compounds and the computational details can be found in the Supplementary Methods, Supplementary Figs. 1–30, and Supplementary Tables 1–5. Cartesian coordinates are available in Supplementary Data 1.

### Catalytic reaction

A typical experimental procedure for the catalytic reactions is described below. To a mixture of [Mo(N)I(PNP)] (**1**) (6.3 mg, 0.010 mmol) and SmI₂ (198 mg, 0.36 mmol) was added THF (3 mL) under N₂. Then, a THF solution (2 mL) containing phenyl chloroformate (18.8 mg, 0.12 mmol) was added, and the mixture was stirred at room temperature for 16 h. After the reaction, volatiles were removed in vacuo. A KOH aqueous

**Table 1 | Molybdenum-catalysed formation of NCO⁻ from reaction of $N_2$ with $SmI_2$ and chloroformates under ambient reaction conditions**

| run | cat. | R of ROCOCl | NCO⁻ equiv/Mo |
|---|---|---|---|
| 1 | **1** | Ph | 0.45 |
| 2 | **5** | Ph | 0.24 |
| 3 | **6** | Ph | 0.74 |
| 4[a] | **6** | Ph | $1.43 \pm 0.12$[b] |
| 5[a] | **6** | Me | $2.51 \pm 0.20$[b] |
| 6[a] | **6** | Et | 0.93 |
| 7[a] | **6** | iPr | 0.54 |
| 8[a] | – | Me | 0.0 |
| 9[a,c] | **6** | Me | 0.18 |
| 10[a] | **6** | – | 0.0 |
| 11[a,d] | **6** | Me | 0.79 |
| 12[e] | **6** | Me | $8.99 \pm 0.15$[b] |

To a mixture of Mo catalyst (0.01 mmol) and $SmI_2$ (0.36 mmol, 36 equiv to Mo) in THF (5 mL) was added a chloroformate ester (0.12 mmol, 12 equiv to Mo) at room temperature and the mixture was stirred at room temperature under 1 atm of $N_2$ (total 16 h).

[a]Chloroformate ester was added via syringe pump over a period of 3 h and the reaction mixture was stirred for an additional 13 h (total 16 h).
[b]Data were mean of multiple individual experiments (at least 2) with error bars representing s.d.
[c]Without $SmI_2(thf)_2$.
[d]Under Ar atmosphere.
[e]Chloroformate ester (0.24 mmol) was added via syringe pump over a period of 12 h and the reaction mixture was stirred for additional 4 h (total 16 h).

solution (0.1 M, 25 mL) was added to the residual solid, and the mixture was stirred at room temperature for 30 min. The supernatant of the suspension was filtered through OnGuard II Na. The yield of NCO⁻ in the residual solution was determined by ion chromatography.

## Data availability

The X-ray crystallographic coordinates for structures reported in this article have been deposited at the Cambridge Crystallographic Data Centre (CCDC), under deposition number CCDC 2159843 ([**1**][I]), 2159844 (**2**), 2159845 (**3a**), 2159846 (**3b·n**CH₃CN), 2159847 (**4a**), 2159848 (**4b**) and 2159849 ([Mo(NCO)Cl(PNP)]₂(μ-N₂)). These data can be obtained free of charge from The Cambridge Crystallographic Data Centre via www.ccdc.cam.ac.uk/ data_request/cif. Cartesian coordinates are available in Supplementary Data 1. All other data are available from the authors upon request.

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

## Acknowledgements

The present project is supported by CREST, JST (JPMJCR1541). We thank Grants-in-Aids for Scientific Research (Nos. JP20H05671, 20K21203, and 22K19041) from JSPS. T.I. is a recipient of the JSPS Predoctoral Fellowships for Young Scientists. This paper is based on results obtained from a project, JPNP21020, commissioned by the New Energy and Industrial Technology Development Organization (NEDO).

## Author contributions

Y.N. and K.Y. conceived and designed this project. T.I., K.A., K.S., S.S. and S.K. conducted the experimental work, including the X-ray analysis. A.E. and H.T. conducted the theoretical studies. All authors discussed the results and drafted the manuscript.

## Competing interests

The authors declare no competing interests.
