## [Peer Review File · Nature Communications]

Direct synthesis of cyanate anion from dinitrogen catalysed by molybdenum complexes bearing pincer-type ligandEditorial Note:

This manuscript has been previously reviewed at another journal that is not operating a transparent peer review scheme. This document only contains reviewer comments and rebuttal letters for versions considered at Nature Communications.

REVIEWERS' COMMENTS

Reviewer #1 (Remarks to the Author):

The manuscript reports catalytic reduction of dinitrogen to isocyanate. It is an impressive accomplishment, although a syringe pump is still needed. As this work sets a precedent for catalysis in this transformation that to date had only been accomplished in stoichiometric reactions, the work is appropriate for publication in Nature Communications. The revisions have appropriately addressed all reviewer comments from the prior submission.

Reviewer #2 (Remarks to the Author):

The authors have fully and satisfactorily addressed my technical concerns . I believe this work now makes a really nice contributions and I highly recommend publication in Nature Communication. I am convinced this will be an highly cited work as it shows that catalytic production of cyanate from dinitrogen may be in reach.

Point-by-Point Response to the Referees.

As for the comments by Referee 1

Referee 1 pointed out that “The manuscript reports catalytic reduction of dinitrogen to isocyanate. It is an impressive accomplishment, although a syringe pump is still needed. As this work sets a precedent for catalysis in this transformation that to date had only been accomplished in stoichiometric reactions, the work is appropriate for publication in *Nature Communications*. The revisions have appropriately addressed all reviewer comments from the prior submission.” As shown here, Referee 1 recommended the publication of our previous version without further revision. Thank you very much for the recommendation for the publication in *Nature Communications*.

As for the comments by Referee 2

Referee 2 pointed out that “The authors have fully and satisfactorily addressed my technical concerns. I believe this work now makes a really nice contributions and I highly recommend publication in *Nature Communication*. I am convinced this will be a highly cited work as it shows that catalytic production of cyanate from dinitrogen may be in reach.” As shown here, Referee 2 recommended the publication of our previous version without further revision. Thank you very much for the recommendation for the publication in *Nature Communications*.